# Blood Cell Responses Following Heavy Alcohol Consumption Coincide with Changes in Acute Phase Reactants of Inflammation, Indices of Hemolysis and Immune Responses to Ethanol Metabolites

**DOI:** 10.3390/ijms232112738

**Published:** 2022-10-22

**Authors:** Onni Niemelä, Anni S. Halkola, Aini Bloigu, Risto Bloigu, Ulla Nivukoski, Heidi Pohjasniemi, Johanna Kultti

**Affiliations:** 1Department of Laboratory Medicine and Medical Research Unit, Seinäjoki Central Hospital, Hanneksenrinne 7, 60220 Seinäjoki, Finland; 2Faculty of Medicine and Health Technology, Tampere University, 33520 Tampere, Finland; 3Center for Life Course Health Research, University of Oulu, 90220 Oulu, Finland; 4Infrastructure of Population Studies, Faculty of Medicine, University of Oulu, 90220 Oulu, Finland

**Keywords:** autoantibodies, calprotectin, erythrocyte, ethanol, haptoglobin, hemolysis, leukocyte, neutrophil, oxidative stress

## Abstract

Aberrations in blood cells are common among heavy alcohol drinkers. In order to shed further light on such responses, we compared blood cell status with markers of hemolysis, mediators of inflammation and immune responses to ethanol metabolites in alcohol-dependent patients at the time of admission for detoxification and after abstinence. Blood cell counts, indices of hemolysis (LDH, haptoglobin, bilirubin), calprotectin (a marker of neutrophil activation), suPAR, CD163, pro- and anti-inflammatory cytokines and autoantibodies against protein adducts with acetaldehyde, the first metabolite of ethanol, were measured from alcohol-dependent patients (73 men, 26 women, mean age 43.8 ± 10.4 years) at baseline and after 8 ± 1 days of abstinence. The assessments also included information on the quantities of alcohol drinking and assays for biomarkers of alcohol consumption (CDT), liver function (AST, ALT, ALP, GGT) and acute phase reactants of inflammation. At baseline, the patients showed elevated values of CDT and biomarkers of liver status, which decreased significantly during abstinence. A significant decrease also occurred in LDH, bilirubin, CD163 and IgA and IgM antibodies against acetaldehyde adducts, whereas a significant increase was noted in blood leukocytes, platelets, MCV and suPAR levels. The changes in blood leukocytes correlated with those in serum calprotectin (*p* < 0.001), haptoglobin (*p* < 0.001), IL-6 (*p* < 0.02) and suPAR (*p* < 0.02). The changes in MCV correlated with those in LDH (*p* < 0.02), MCH (*p* < 0.01), bilirubin (*p* < 0.001) and anti-adduct IgG (*p* < 0.01). The data indicates that ethanol-induced changes in blood leukocytes are related with acute phase reactants of inflammation and release of neutrophil calprotectin. The studies also highlight the role of hemolysis and immune responses to ethanol metabolites underlying erythrocyte abnormalities in alcohol abusers.

## 1. Introduction

Excessive ethanol consumption is a common cause of abnormalities in both blood cell counts and morphology [1,2]. In erythrocytes, the most typical aberrations are elevated mean corpuscular volume (MCV) and mean corpuscular hemoglobin (MCH), the former being alcohol-related, even in two thirds of consecutive adult patients with macrocytosis [2,3,4]. Recent studies have further indicated that excessive alcohol drinking may also interfere with the regulation of leukocyte function [5,6,7].

While the primary mechanisms of the ethanol-induced effects on blood cells have remained poorly understood, previous studies on red cells have suggested that alcohol drinking may induce structural adaptations on erythrocyte cell membranes [8]. The membrane interactions with reactive ethanol metabolites or lipopolysaccharide (LPS) may lead to the generation of adducts with proteins and cellular constituents and induce immune responses towards such neoantigens [9,10,11,12,13,14,15,16,17,18,19]. In addition, alcohol intake may affect pathways of leukocyte regulation together with stimulation of systemic inflammation and oxidative stress [5,6,7,20,21,22,23]. The status of inflammation in alcoholic patients is frequently compromised and could be influenced by both the amounts of drinking and the presence or absence of tissue injury [20,23,24,25,26,27,28,29].

To date, only limited information has, however, been available on simultaneous biomarker-based comparisons of the impacts of ethanol consumption on blood cell indices and leukocyte recruitment. The aim of this work was to examine the status of blood cells, indices of hemolysis, autoantibodies against ethanol metabolites and pro- and anti-inflammatory mediators of inflammation in consecutive alcohol-dependent patients admitted for detoxification. The biomarker levels at baseline and after supervised abstinence were also compared with the data obtained from markers of alcohol consumption and liver status. Our findings reveal distinct interactions between such parameters, which may prove to be of value in studies on the pathogenic mechanisms of ethanol-induced abnormalities in blood cells.

## 2. Results

Table 1 summarizes the data on the main blood cell findings, hemolytic parameters, markers of alcohol consumption and liver status and mediators of inflammation and autoantibody levels in the alcohol-dependent patients at the time of admission and after supervised abstinence. At baseline, patients showed elevated values in all biomarkers of alcohol consumption and liver status, and a significant decrease towards the normal range was found upon abstinence. For blood cell indices, the period of supervised abstinence was, however, found to result in a significant increase in blood leukocyte levels (*p* < 0.001), the mean corpuscular volume of erythrocytes (MCV) (*p* < 0.02) and platelet numbers (*p* < 0.02). In hemolytic indices, lactate dehydrogenase (LDH) at baseline was above the reference range in 39% and haptoglobin in 22% of the study subjects and during the follow-up significant decreases were found to occur in LDH activities (*p* < 0.001) and in serum bilirubin levels (*p* < 0.001).

Among the various mediators of inflammation, a significant decrease during the follow-up period was noted only in the level of CD163, a scavenger receptor for haptoglobin-hemoglobin complexes (*p* < 0.01), whereas serum suPAR, a soluble urokinase plasminogen activator receptor, showed a significant increase (*p* < 0.01). Serum calprotectin, interleukin-6 (IL-6), IL-8, tumor necrosis factor-alpha (TNF-α) and IL-10 remained relatively stable during this period of follow-up. The antibodies representing IgM and IgA isotypes against acetaldehyde-derived protein modifications were found to decrease during the period of abstinence, whereas IgG titers remained unchanged.

Table 2 summarizes the correlations between the various study parameters. Blood leukocyte levels at baseline showed significant positive correlations with serum calprotectin (r_s_ = 0.610, *p* < 0.001), haptoglobin, (r_s_ = 0.314, *p* < 0.01) and IL-6 (r_s_ = 0.363, *p* < 0.001), suPAR (r_s_ = 0.315, *p* < 0.01), and a significant inverse correlation with IgG antibodies (r_s_ = −0.551, *p* < 0.001) against the ethanol-derived protein modifications (Table 2). Significant positive correlations at baseline were observed with MCV and serum LDH (r_s_ = 0.217, *p* < 0.05), bilirubin (r_s_ = 0.229, *p* < 0.05), aspartate aminotransferase (AST) (r_s_ = 0.308, *p* < 0.01), IL-6 (r_s_ = 0.270, *p* < 0.05), IL-8 (r_s_ = 0.411, *p* < 0.001), suPAR (r_s_ = 0.387, *p* < 0.001) and ferritin, (r_s_ = 0.465, *p* < 0.001). A significant inverse correlation was noted with MCV and IgG (r_s_ = −0.510, *p* < 0.001) and IgM (r_s_ = −0.418, *p* < 0.001) antibodies against ethanol-derived protein modifications (Table 2). The anti-adduct IgG titers also showed a negative correlation between blood hemoglobin (r_s_ = −0.278, *p* < 0.05) and platelet levels (r_s_ = −0.275, *p* < 0.05). Blood hemoglobin levels were positively associated with serum ferritin (r_s_ = 0.311, *p* < 0.01) and urate levels (r_s_ = 0.408, *p* < 0.001).

In the analyses of correlations observed between the magnitude and direction of changes in the blood cell indices during the follow-up, the changes in blood leukocyte levels were found to parallel the changes in serum calprotectin (r = 0.484, *p* < 0.001), haptoglobin (r = 0.381, *p* < 0.001), IL-6 (r = 0.326, *p* < 0.02) and suPAR (r = 0.332, *p* < 0.02). The changes in MCV correlated with those in LDH (r = −0.247, *p* < 0.02), MCH (r = 0.284, *p* = 0.01), bilirubin (r = −0.324, *p* < 0.001) and anti-adduct IgG (r = −0.492, *p* < 0.01) (Table 2).

The levels of ethanol consumption from the past one month prior to blood sampling correlated significantly with serum gamma glutamyl transferase (GGT) (r_s_ = 0.329, *p* < 0.01), AST (r_s_ = 0.346, *p* < 0.01), alanine aminotransferase (ALT) (r_s_ = 0.269, *p* < 0.05), MCV (r_s_ = 0.290, *p* < 0.01), serum ferritin (r_s_ = 0.373, *p* < 0.01), haptoglobin (r_s_ = 0.333, *p* < 0.01), CD163 (r_s_ = 0.377, *p* < 0.01), TNF-α (r_s_ = 0.473, *p* < 0.001) and anti-adduct IgA levels (r_s_ = 0.260, *p* < 0.01) (Table 2).

## 3. Discussion

Our study comparing blood cell changes induced by heavy alcohol drinking with various mediators of inflammation, indices of hemolysis and autoantibody responses against ethanol metabolites reveals distinct coinciding phenomena in the responses of such parameters. The observed changes in blood leukocytes, a cellular biomarker of inflammation, were found to parallel those in serum calprotectin, a marker of neutrophil activation, indicating a potent role of alcohol abuse as a driver of neutrophil-driven inflammation [30,31,32,33,34]. Thus, neutrophil function may play a key role in perpetuating inflammation in patients with recent alcohol drinking, which is also in accordance with recent findings from both healthy volunteers and alcoholic patients, indicating that even a single alcohol binge may influence neutrophil recruitment and gut microbiome composition and thereby lead to an increase in acute phase proteins and low-grade inflammation [5,6,7]. Calprotectin release upon neutrophil activation may be associated with both anti-infective and anti-inflammatory properties, including control of myelopoiesis, scavenging of reactive oxygen species, chemotaxis and direct antimicrobial effects [30,32,33,34,35,36]. The interactions observed here with blood leukocytes, calprotectin and the other mediators of inflammation also support a role of such responses in controlling the status of inflammation in response to excessive alcohol drinking [32,33,35,36,37]. The leukocyte responses were also associated with the levels of haptoglobin, a major hemoglobin-binding protein in the plasma, which is known as an acute phase protein with the ability to regulate immune cell responses [38].

The present findings from red blood cells showed that high red cell size (MCV), which has been previously recognized as the most typical abnormality in blood cell examinations from alcohol-consuming patients, is frequently elevated in alcohol-dependent patients and similar to the response in blood leukocytes, and increased even further after the period of abstinence. MCV levels at baseline correlated with indices of hemolysis, LDH and bilirubin, supporting the view that although haptoglobin levels were not found to be depleted here, the erythrocytes of alcoholics may be prone to hemolysis after bouts of heavy drinking. The biomarkers of alcohol consumption, IL-6 and IL-8 cytokines and serum ferritin, an acute phase reactant of inflammation, also correlated with MCV at baseline. The correlation observed between MCV and IgG antibodies towards acetaldehyde adducts suggests that immune responses directed against the ethanol-derived modifications in erythrocytes may also play a role in blood cell responses in alcoholics [13,14,15,16]. Erythrocytes have previously been suggested to be targets of acetaldehyde adduct formation in vivo [15] and capable of serving as bioreactors for removal of the toxic ethanol metabolites [11]. Thus, it is possible that antigen-driven humoral immune responses could also contribute to exclusion and neutralization of ethanol-derived neoantigens. Upon overwhelming antigenic stimulation the formation of immune complexes could, however, cause monocytes to release mediators of inflammation and tissue damage. Previous studies have shown that immune responses raised through intestinally induced B-cells from the ethanol-exposed epithelial tissues may also play a role in alcohol-induced liver or kidney damage [10,14,26,39,40]. The gastrointestinal tract is rich in enzymes capable of metabolizing ethanol to acetaldehyde, and previously, elevated MCV levels have also been linked with a risk for carcinogenesis in alcoholics with high gastrointestinal levels of acetaldehyde [41,42,43].

The responses in suPAR and CD163, an endocytic receptor protein for haptoglobin-hemoglobin complexes and a biomarker of macrophage activation, were also found to be associated with the changes in blood cells, indices of hemolysis and the status of inflammation. Previously, suPAR and CD163 proteins have both emerged as predictive risk markers in inflammatory conditions [44,45,46,47,48,49,50]. suPAR, which showed here an increase during the period of abstinence, is expressed on neutrophils, monocytes and activated T-cells [44,48,50] whereas CD163, which decreased during abstinence, is present on macrophages and monocytes. Its serum levels are known to become elevated in conjunction with excessive activation of macrophages [45,49]. It may also play a role in innate immune defense by sequestering hemoglobin-bound iron and scavenging of oxidative stress -induced by-products [45,51].

The changes in the inflammatory status in alcoholics also appears to be a major determinant in the sequence of events stemming from heavy alcohol drinking to tissue injury [26]. In line with this view, increased levels of circulating neutrophils in alcoholics seem to be associated with increased serum liver enzyme activities and upregulation of pro-inflammatory cytokines [6,52,53]. With continuing heavy alcohol drinking, the balance between pro- and anti-inflammatory cytokines may become skewed with overwhelming production of pro-inflammatory mediators, which has previously been observed in patients with alcoholic liver disease [25,26,54]. The molecular mechanisms responsible for causing either over-activation or downregulation of leukocyte activation have, however, remained poorly defined. This could result from excessive antigen loading due to ethanol metabolites, gut-derived endotoxins and bacterial products as well as impaired anti-inflammatory capacity, triggering the production of reactive oxygen species and oxidative stress [40,54,55,56,57,58,59]. Endotoxins have also been suggested to be able to bind to erythrocyte membranes inducing hemolysis and ineffective erythropoiesis [9]. Interestingly, alkaline phosphatase (ALP), which in turn has the ability to detoxify endotoxins [60], was found here to correlate with the indices of hemolysis (LDH, bilirubin), MCV, MCH, leukocytes, IL6, CD 163 and calprotectin.

A switch of the immune system towards a pro-inflammatory direction may also be influenced by additional precipitating factors, such as the metabolic status and intracellular pH [35]. In turn, IL-6 and IL-10 cytokine responses can activate anti-inflammatory cascades by stimulating Th2 and inhibiting TNF-α [59,61], a potent pro-inflammatory mediator of tissue damage and oxidative stress [59,62,63,64], which in the present series was also found to correlate with the amounts of recent drinking. TNF-α has the ability to attract neutrophils and regulate macrophage production [59,65]. The pro-inflammatory cytokine, IL-8, also shows that distinct target specificity for neutrophils and its enhanced expression in experimental models coincides with decreased hepatocyte survival [66].

Current follow-up data further reveals that while upon abstinence, there is a significant decrease in biomarkers of alcohol consumption and liver cell damage, many of the mediators of inflammation remain elevated and stable at least during this relatively short period of abstinence, whereas the leukocyte levels and suPAR levels continue to increase. Interestingly, previous studies have indicated that clinical deterioration, which is frequently observed among alcohol-dependent patients for several days following hospitalization and cessation of ethanol intake, may be mediated by immunological mechanisms [67]. Some of the responses in immune status and mediators of inflammation following chronic alcohol drinking may also be expected to take place in an alcohol-dose dependent manner [29]. An excess of pro-inflammatory mediators can maintain the unfavorable over-activation of the sympathetic nervous system, oxidative stress and the immunocompromised status for lengthy periods following heavy alcohol intake [23,55,56,68]. This could also play a role in the individual susceptibility to coincidentally occurring pathogens in alcoholic patients during this period [69,70]. Taken together, the present findings among heavy alcohol drinkers demonstrate distinct features in blood leukocyte and erythrocyte status, which coincide with changes in markers of neutrophil activation, indices of hemolysis, mediators of inflammation and immune responses towards ethanol metabolites. The sample size in our study was relatively small although large enough for new findings to emerge. Future work, however, warrants addressing the pathogenic and clinical significance of current observations on hematotoxicity, susceptibility to infections as well as on the pathways of low-grade inflammation and tissue damage in heavy alcohol drinkers [2,13,23,32]. Additional research is also needed to address the question whether modulation of immunological responses could be used to develop new approaches in the treatment of alcohol-induced inflammatory manifestations. It should further be noted that due to the small sample size, it is possible that our findings are not necessarily generalizable to all patients with alcohol use disorders. The present sample was derived from a society representing a high prevalence of binge drinking, which based on previous observations seem to be at a higher risk for developing health problems than those with regular alcohol consumption [71].

## 4. Materials and Methods

### 4.1. Participants

The present study population comprised of 99 alcohol-dependent patients who had been admitted for detoxification. The mean age was 43.8 ± 10.4 years and there were 73 men and 26 women. Blood samples were collected at the time of admission and following an 8 ± 1 day period of supervised abstinence. All subjects were devoid of clinical and laboratory records of significant liver disease, comorbid substance abuse, major depression, inflammatory bowel diseases or any immunological disorders. The documentation of alcohol use was based on hospital records and detailed clinical interviews using a time-line follow-back technique recording data from previous one month and one week preceding admission. All study subjects showed a history of heavy drinking consisting of continuous alcohol consumption and repeated episodes of binge drinking, the mean (SD) recent alcohol consumption being 105 ± 70 grams/day from the period of one month prior to sampling. The mean duration of abstinence prior to the first sampling time point was 2 ± 1 days. During hospitalization, the study subjects volunteered for a follow-up with supervised abstinence. All patients were hospitalized inpatients over the entire study period. Blood alcohol concentrations during this period were controlled by repeated analyses from breath air. The severity of alcohol withdrawal symptoms were monitored by clinical assessment and treated with symptom control benzodiazepine medication during the first day (day 1 to day 3) after admission, if needed. In addition, supportive care and vitamins were used.

All subjects gave their informed consent for the study. The protocol was approved by the local Southern Ostrobothnia Hospital District and Tampere University Hospital ethical committee, and the study was conducted according to the provisions of the Declaration of Helsinki.

### 4.2. Laboratory Methods

Blood cell counts from fresh blood samples were determined using a Sysmex XE-5000 automated Hematology Analyzer (Sysmex Europe SE). Serum samples were prepared by centrifugation (1500× *g* for 10 min) and routine blood chemistry analyses were carried out using standard clinical chemical methods on an Abbott Architect c8000 automated clinical chemistry analyzer (Abbott Diagnostics, Abbott Laboratories, Abbott Park, IL, USA). Carbohydrate-deficient transferrin (CDT) was measured on a Siemens BN Prospec immunoassay according to the instructions of the manufacturer (Siemens Diagnostics, Erlange, Germany). The measurements of the various mediators of inflammation were carried out from aliquots of serum samples which had been stored frozen at –70 °C prior to the analyses. The concentrations of interleukins (IL-6, IL-8, IL-10) and TNF-α were determined using Quantikine high sensitivity ELISA kits (R&D Systems, Abingdon, Science Park, UK). Serum calprotectin measurements were carried out using the BÜHLMANN MRP8/14 ELISA kit according to the instructions of the manufacturer (BÜHLMANN Laboratories AG, Schönenbuch, Switzerland). Serum suPAR concentrations were measured using the suPARnostic enzyme-linked immunosorbent assay (ELISA) kit (Virogates, Birkerød, Denmark) and CD163 antigen using Quantikine human CD163 ELISA assay (R&D Systems, Abingdon Science Park, UK). All measurements were carried out in a SFS-EN ISO 15189:2013 accredited laboratory.

### 4.3. Measurements of Antibody Titers against Acetaldehyde-Modified Antigens

The preparation of acetaldehyde-modified antigens in vitro was first carried out using human erythrocyte protein prepared from EDTA-blood of a teetotaler, as previously described [26]. In essence, the erythrocytes were separated by centrifugation and washed three times with an equal volume of phosphate-buffered saline (PBS: 7.9 mM Na_2_HPO_4_, 1.5 mM KH_2_PO_4_, 137 mM NaCl, 2.7 mM KCl, pH 7.4), lysed with polyoxyethylene ether, 0.1% *v/v* in borate buffer (Hemolysis Reagent, DIAMAT^TM^ Analyzer System, Bio-Rad), and incubated for 35 min at +37 °C to remove unstable Schiff bases. The hemolysate was brought into a hemoglobin protein concentration of 12 mg/mL with PBS and stored frozen in aliquots at –70 °C prior to use. Acetaldehyde diluted in PBS was added to aliquots of the freshly prepared hemoglobin, containing 12 mg protein/mL, to obtain a final acetaldehyde concentration of 10 mM. The mixture was allowed to react in a tightly sealed container at +4 °C overnight. Protein adducts were reduced by addition of sodium cyanoborohydride (10 mM) and mixing for 5 h at +4 °C. All protein solutions were dialyzed twice against PBS at +4 °C and stored in small aliquots for single use at –70 °C. Samples representing unmodified protein were prepared and treated similarly to that of the modified protein except for the addition of acetaldehyde.

For measuring the antibody titers, microtiter plates (Nunc-Immuno Plate, Maxisorb^TM^, InterMed, Denmark) were coated with acetaldehyde-modified hemoglobin, or corresponding unmodified proteins (background) in PBS (3 µg protein in 100 µL/well) and incubated for 1½ h at +37 °C. Nonspecific binding was blocked by incubation with 0.2% gelatin in PBS (150 µL/well) for 1 h at +37 °C. The sample sera were diluted (1:40) in PBS, which contained 0.04% Tween-20 (PBS-Tween). The final volume of 50 µL of each serum dilution were allowed to react with the coated proteins for 1 h at +37 °C followed by extensive washing with PBS-Tween. Alkaline phosphatase -linked goat anti-human immunoglobulins IgA, IgG, or IgM (Jackson ImmunoResearch Laboratories, Inc., West Grove) were used to label antibody-antigen complexes (50 µL/well). The immunoglobulins were diluted in PBS-Tween containing 8 mM MgCl_2_ and a small amount of dithiothreitol (DTT). The plates were allowed to incubate at +4 °C overnight. After washing, 100 µL of p-nitrophenylphosphate-solution was added for the color reaction substrate (Alkaline Phosphatase Substrate Kit, Bio-Rad Laboratories, Hercules, CA). Color reactions were stopped by adding 100 µL of 0.4 M NaOH, and the optical densities were read at 405 nm by Anthos HTII microplate reader (Anthos Labtec Instruments, Salzburg, Austria).

### 4.4. Statistical Methods

The values are reported as mean ± standard deviation (SD) for normally distributed variables or medians and interquartile ranges (IQR) for skewed variables, as indicated. The comparisons between the changes in the study parameters at different time points were made using paired samples *t*-test since the changes between the measures showed adequately normal distribution. Due to the skewed variables, the correlations between the laboratory measures at admission were evaluated by Spearman’s rank correlation coefficient (r_s_). The relative changes at measures (calculated as absolute changes divided by the values at admission) followed normal distribution and the corresponding correlations were evaluated by Pearson’s correlation coefficient (r). A *p*-value of < 0.05 was considered statistically significant. Statistical analyses were carried out using IBM SPSS Statistics 28.0 (Armonk, NY, USA: IBM Corp.).

## Figures and Tables

**Table 1 ijms-23-12738-t001:** Laboratory markers in alcoholics at the time of admission (baseline) and in follow-up samples after 8 ± 1 days of supervised abstinence.

Parameter	Unit	Baseline Mean (SD)	Follow-Up Mean (SD)	Direction of Change	*p **
Blood cell counts					
Leukocytes	10^9^/L	7.50 (2.26)	8.30 (2.89)	↑	<0.001
Hemoglobin	g/L	148.2 (12.2)	147.4 (10.7)		
MCV	fL	95.7 (5.2)	96.5 (5.1)	↑	<0.02
MCH	pg	31.8 (2.0)	31.7 (2.0)		
Thrombocytes	10^9^/L	215 (86)	287 (105)	↑	<0.02
Hemolytic markers					
Haptoglobin	g/L	1.61 (0.60)	1.65 (0.69)		
LDH	U/L	224 (60)	190 (37)	↓	<0.001
Bilirubin	µmol/L	12.9 (18.1)	7.9 (10.6)	↓	<0.001
Alcohol markers and liver status					
CDT	%	2.80 (1.51)	1.70 (0.54)	↓	<0.001
GGT	U/L	188.7 (360.8)	83.4 (102.3)	↓	<0.005
AST	U/L	57.9 (48.4)	29.7 (16.5)	↓	<0.001
ALT	U/L	60.5 (59.8)	43.0 (41.8)	↓	<0.005
ALP	U/L	97.2 (52.6)	85.6 (47.8)	↓	<0.01
Acute phase reactants of inflammation					
IL-6	pg/mL	4.7 (3.5)	4.0 (3.9)		
IL-8	pg/mL	33 (21)	29 (17)		
IL-10	pg/mL	0.8 (0.4)	0.8 (0.2)		
TNF-α	pg/mL	1.3 (0.3)	1.4 (0.3)		
suPAR	ng/mL	3.5 (1.2)	3.9 (1.5)	↑	0.007
CD163	ng/mL	854 (388)	735 (351)	↓	0.010
Calprotectin	µg/mL	4.6 (3.2)	4.8 (3.2)		
Autoantibodies					
Anti Ach adduct IgA	U/L	125 (128)	97 (111)	↓	<0.02
Anti Ach adduct IgG	U/L	355 (140)	360 (142)		
Anti Ach adduct IgM	U/L	670 (94)	650 (118)	↓	<0.05

* paired samples *t*-test, ALT, alanine aminotransferase; AST, aspartate aminotransferase; ALP, alkaline phosphatase; CD163, a biomarker of monocyte-macrophage activation; CDT, carbohydrate-deficient transferrin; IL, interleukin; Ig, immunoglobulin; GGT, gamma glutamyl transferase; LDH, lactate dehydrogenase; MCH, mean corpuscular hemoglobin; MCV, mean corpuscular volume; suPAR, soluble urokinase plasminogen activator receptor, a biomarker of immune activation; TNF-α, tumor necrosis factor-alpha. The arrows indicate the direction of change in the biomarker levels during abstinence (**↑** increase; **↓** decrease).

**Table 2 ijms-23-12738-t002:** Correlations between the study parameters.

	Leukocytes	Hemoglobin	MCV	MCH	Haptoglobin	LDH	Bilirubin	CDT	GGT	AST	ALT	ALP	IL-6	IL-8	IL-10	TNF-α	suPAR	CD163	Calprotectin	Ferritin	Urate	Anti Ach adduct IgA	Anti Ach adduct IgG	Anti Ach adduct IgM	Consumption/ previous month
Leukocytes	1.000																								
Hemoglobin	0.116	1.000																							
MCV	0.074	−0.044	1.000																						
MCH	−0.034	0.179	0.854	1.000																					
Haptoglobin	0.314	0.050	0.021	−0.004	1.000																				
LDH	0.044	0.077	0.217	0.277	0.050	1.000																			
Bilirubin	−0.111	0.156	0.229	0.384	−0.061	0.416	1.000																		
CDT	0.039	−0.063	0.133	0.123	−0.182	0.079	0.117	1.000																	
GGT	−0.088	0.119	0.378	0.408	0.171	0.535	0.445	−0.018	1.000																
AST	−0.069	0.157	0.308	0.353	0.054	0.662	0.551	0.236	0.738	1.000															
ALT	0.060	0.258	0.188	0.273	0.107	0.498	0.336	0.160	0.614	0.764	1.000														
ALP	0.217	0.034	0.206	0.220	0.116	0.463	0.301	−0.030	0.438	0.489	0.404	1.000													
IL-6	0.363	−0.047	0.270	0.216	0.342	0.285	0.303	−0.085	0.348	0.331	0.123	0.369	1.000												
IL-8	−0.107	0.019	0.411	0.323	0.049	0.382	0.327	0.192	0.428	0.566	0.502	0.432	0.299	1.000											
IL-10	0.069	−0.039	−0.225	−0.202	−0.114	−0.059	0.274	0.000	−0.076	−0.049	−0.237	0.148	0.084	−0.316	1.000										
TNF-α	−0.167	−0.116	−0.172	0.061	0.337	0.019	0.323	0.111	0.124	0.312	0.176	0.080	0.395	0.422	0.671	1.000									
suPAR	0.315	0.027	0.387	0.272	0.076	0.288	0.037	0.108	0.299	0.420	0.301	0.264	0.359	0.448	−0.047	0.226	1.000								
CD163	−0.062	0.202	0.259	0.206	0.006	0.459	0.350	0.190	0.485	0.688	0.558	0.361	0.243	0.411	−0.081	0.195	0.500	1.000							
Calprotectin	0.610	0.301	0.193	0.198	0.186	0.290	0.149	0.214	0.075	0.125	0.250	0.315	0.236	0.347	0.269	0.014	0.490	0.175	1.000						
Ferritin	−0.070	0.311	0.465	0.514	−0.018	0.581	0.484	0.057	0.722	0.745	0.714	0.487	0.126	0.516	−0.206	0.119	0.372	0.611	0.196	1.000					
Urate	0.134	0.408	0.135	0.170	−0.034	0.017	0.374	−0.069	0.224	0.154	0.174	−0.051	0.250	0.219	0.439	0.089	−0.021	0.043	−0.044	0.077	1.000				
Anti Ach adduct IgA	0.072	0.105	−0.100	−0.078	−0.181	−0.092	0.332	0.360	−0.209	−0.047	−0.111	0.003	0.301	−0.214	−	0.090	−0.104	0.156	0.327	0.053	0.176	1.000			
Anti Ach adduct IgG	−0.551	−0.278	−0.510	−0.467	−0.109	−0.191	−0.107	−0.119	−0.262	−0.155	−0.195	−0.077	−0.237	−0.103	−	0.154	−0.263	−0.135	−0.352	−0.209	−0.056	0.203	1.000		
Anti Ach adduct IgM	−0.231	−0.109	−0.418	−0.308	−0.350	−0.277	−0.078	−0.168	−0.466	−0.357	−0.564	−0.023	−0.104	−0.085	−	0.165	−0.371	−0.554	−0.273	−0.357	−0.041	0.072	0.429	1.000	
Consumption/previous month	−0.099	−0.053	0.290	0.228	0.333	0.072	0.275	0.251	0.329	0.346	0.269	0.160	0.281	0.105	−0.158	0.473	0.169	0.377	−0.015	0.373	0.053	0.260	−0.236	−0.176	1.000

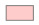
*p* < 0.05, 


*p* < 0.01. For abbreviations, see Table 1.

## Data Availability

The datasets generated during the current study are not publicly available due to restrictions relating to confidential patient information but are available from the corresponding author on reasonable request.

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
