# Peer review of "Blood Cell Responses Following Heavy Alcohol Consumption Coincide with Changes in Acute Phase Reactants of Inflammation, Indices of Hemolysis and Immune Responses to Ethanol Metabolites"

_ijms, 2022, doi:10.3390/ijms232112738_

Round 1

Reviewer 1 Report

This is a well written study that provides information that has some novelty. Despite that, in its current form, the manuscript has some shortcomings that should be noted.

The flow of ideas in the introduction need attention and the information provided is very basic.

The number of patients included in the study is on the lower side. Despite that the authors also analyze levels after more than a week of abstinence, the statistical analysis are crude and seem like a fishing expedition. Therefore, the authors should note that the study is, at its best, exploratory in nature.

Author Response

This is a well written study that provides information that has some novelty. Despite that, in its current form, the manuscript has some shortcomings that should be noted.

The flow of ideas in the introduction need attention and the information provided is very basic.

We thank the Reviewer for the positive overall assessment and useful suggestions concerning our work. Based on the recommendation given, we have now improved the clarity of the presentation in the Introduction Section by summarizing the existing research and positioning our own approach in a more concise manner.

The number of patients included in the study is on the lower side. Despite that the authors also analyze levels after more than a week of abstinence, the statistical analysis are crude and seem like a fishing expedition. Therefore, the authors should note that the study is, at its best, exploratory in nature.

We agree with the view that a larger number of subjects could have further increased the statistical power in the present study. We feel, however, that the total number of patients (n=99), use of follow-up data and analyses of a wide variety of laboratory tests should provide a strong basis for a comprehensive assessment of the biomarker status in heavy drinkers at the time of admission and after the first days of abstinence. Based on the reviewer`s suggestion, we have also strengthened the Discussion by emphasizing the relatively small number of participants as a possible limitation of the study.

Reviewer 2 Report

The manuscript titled: Blood cell responses following heavy alcohol consumption coincide with changes in acute phase reactants of inflammation, indices of hemolysis and immune reactions to ethanol metabolites, by Niemela O. et al. has the aim to The aim of this work was to examine the status of blood cells, indices of hemolysis, autoantibodies against ethanol metabolites and both pro- and anti-inflammatory mediators of inflammation in consecutive alcohol-dependent patients who had been admitted for detoxification.
Conclusions suggest that among heavy alcohol drinkers demonstrate present distinct features in blood leukocyte and erythrocyte status, associated with changes in markers of neutrophil activation, indices of hemolysis, mediators of inflammation and
immune responses towards ethanol metabolites.

The paper has some problems to be resolved before publication.

Major troubles
1. was detoxification performed in inpatient or outpatient patients?

2. How have withdrawal syndrome (AWS) been treated? Did you consider the possible pharmacological interactions?
3. Have you monitored AWS in the detoxification days? What test did you use for monitoring?
4. Did you have the possibility to indicate How many High-Risk drinkers were in your study and give information if patients as binge drinking or not? Are there some differences between drinking modalities?

Author Response

The manuscript titled: Blood cell responses following heavy alcohol consumption coincide with changes in acute phase reactants of inflammation, indices of hemolysis and immune reactions to ethanol metabolites, by Niemela O. et al. has the aim to The aim of this work was to examine the status of blood cells, indices of hemolysis, autoantibodies against ethanol metabolites and both pro- and anti-inflammatory mediators of inflammation in consecutive alcohol-dependent patients who had been admitted for detoxification.
Conclusions suggest that among heavy alcohol drinkers demonstrate present distinct features in blood leukocyte and erythrocyte status, associated with changes in markers of neutrophil activation, indices of hemolysis, mediators of inflammation and immune responses towards ethanol metabolites.

The paper has some problems to be resolved before publication.

Major troubles
1. Was detoxification performed in inpatient or outpatient patients?

In this material, all patients were hospitalized inpatients over the entire study period. We feel that this approach was useful not only for maintenance of abstinence but also for allowing the opportunity for medical counseling and support during the hospital follow-up. In the revised manuscript, we have now added this information in the Methods section (paragraph: Participants).

  1. How have withdrawal syndrome (AWS) been treated? Did you consider the possible pharmacological interactions?

Based on the severity of the withdrawal symptoms the patients were treated with symptom control medication during the first (day 1 to day 3) after admission, as needed. The medication included primarily benzodiazepins, such as diazepam or chlordiazepoxide with symptom-triggered front-loading. In addition, supportive care and vitamins were used. The baseline samples were obtained before treatment. At the time of collecting the follow-up samples the drug treatment had already been discontinued. Based on the timing of the study samples and previous literature on the possible interactions with benzodiazepins and current biomarkers, we do not expect any significant bias here through pharmacological interactions.  A more detailed account on the above aspects has now been given in the revised manuscript.

  1. Have you monitored AWS in the detoxification days? What test did you use for monitoring?

In the revised manuscript, we have now added information on the assessment of AWS. In this work, alcohol withdrawal was a clinical diagnosis. The severity of the symptoms were monitored by clinical assessments including agitation, paroxysmal sweats, tremor, headache and nausea/vomiting as the major components. Unfortunately, systematic scoring data for all the CIWA-AR components was not available. The most severe cases of withdrawal, such as delirium tremens, were treated elsewhere in specialized emergency units and therefore, no such cases were included in this study.

  1. Did you have the possibility to indicate How many High-Risk drinkers were in your study and give information if patients as binge drinking or not? Are there some differences between drinking modalities?

The reviewer raises an important point. In our study the clinical examinations and interviews on the history of alcohol intake in all patients indicated high-risk drinking, as defined by NIAAA criteria. In this study we did not systematically classify the subjects to subgroups based on their primary patterns of drinking (binge or regular). It should be noted, however, that in Finland the vast majority of the patients admitted for detoxification clinics represent binge drinkers. In the revised manuscript, we have now sharpened the presentation by including discussion on various drinking modalities and by emphasizing the role of binge-type drinking in creating adverse health effects (Discussion Section, last paragraph).

Many thanks for all the above suggestions.

Round 2

Reviewer 1 Report

I do not feel that the authors have addresses all my prior concerns in a proper fashion.

Reviewer 2 Report

The adjunctive pieces of information are adequate to publication